# The Whole Blood Transcriptomic Analysis in Sickle Cell Disease Reveals RUNX3 as a Potential Marker for Vaso-Occlusive Crises

**DOI:** 10.3390/ijms26136338

**Published:** 2025-06-30

**Authors:** Safa Taha, Hawra Abdulwahab, Muna Aljishi, Ameera Sultan, Moiz Bakhiet, Salvatore Spicuglia, Mohamed Belhocine

**Affiliations:** 1Department of Molecular Medicine, Princess Al Jawhara Center for Molecular Medicine, Genetics and Inherited Diseases, College of Medicine and Health Sciences, Arabian Gulf University, Manama P.O. Box 26671, Bahrain; safat@agu.edu.bh (S.T.); munajma@agu.edu.bh (M.A.); ameeraa@agu.edu.bh (A.S.); moiz@agu.edu.bh (M.B.); mohamedb@agu.edu.bh (M.B.); 2Equipe Labélisée Ligue Contre le Cancer, Aix-Marseille University, INSERM, TAGC, UMR1090, 13288 Marseille, France; salvatore.spicuglia@inserm.fr

**Keywords:** VOC, SCD, microarray, hemolysis, inflammation, biomarker, blood disease, pain

## Abstract

Sickle cell disease (SCD) is the most common hemoglobinopathy, caused by a mutation in the β-globin gene of hemoglobin. It predisposes patients to painful Vaso-occlusive crises (VOC) and multi-organ dysfunctions. The disease exhibits significant phenotypic variability, making it challenging to predict severity and outcomes. This study aimed to characterize the whole blood gene expression profile of Bahraini SCD patients, identifying differentially expressed genes during steady-state (*n* = 10) and VOC (*n* = 10) compared to healthy controls (*n* = 8). Analysis revealed 2073 and 3363 dysregulated genes during steady-state and VOC, respectively, compared to controls, with 1078 genes differentially expressed during VOC versus steady-state. Gene Ontology (GO) enrichment analysis highlighted significant deregulation in immune and hematopoietic pathways, including down-regulation of critical genes for immune modulation and hematopoietic balance. Notably, the transcription factor RUNX3, involved in immune cell differentiation and inflammation, was among the 668 down-regulated genes. RUNX3 was four-fold down-regulated in microarray analysis, three-fold in PCR, and showed a mean protein concentration of 11.13 pg/mL during VOC compared to 457.93 pg/mL during steady-state (*p* < 0.01). These findings suggest that RUNX3 may serve as a potential biomarker for VOC. Future large-scale validation, additional proteomic studies, and functional investigations are recommended to confirm its clinical utility and significance.

## 1. Introduction

Sickle cell disease (SCD) is caused by a mutation in the hemoglobin genes, resulting in the formation of hemoglobin S (HbS) predisposes to hemolytic anemia, painful attacks, and multiple end-organ complications [1,2]. It was the first identified genetic disease [1], and it is the most common hemoglobinopathy in the Kingdom of Bahrain. The earliest presentation of SCD is the acute painful vaso-occlusive crisis (VOC). VOC results from a complex mechanism involving adhesive interactions among sickle cells, leukocytes, endothelium, other blood cells, and plasma factors [3]. Microvascular occlusions occur through the aggregation of red blood cells, leukocytes, and platelets, predisposing them to inflammation and intravascular and extravascular hemolysis [4]. There is a continuous interaction between hemostatic and inflammatory systems in SCD, which further disseminates the hypercoagulability and inflammatory changes [5], as it is well established that innate immune cells contribute to the formation of inflammation, adhesion, and painful attacks [6].

Moreover, SCD patients have increased susceptibility to infection with encapsulated bacteria such as Streptococcus Pneumoniae, Hemophilus Influenzae, and Salmonella due to splenic infarction, together with recurrent VOC, resulting in multiple organ infarctions and end-organ damage leading to various complications [7]. In general, complications can occur in any tissue, such as acute chest syndrome (ACS), pneumonia, stroke, splenic infarction, avascular necrosis of bone, and osteomyelitis [7]. However, some individuals develop certain complications while others do not, regardless of the number of VOC and the degree of anemia [8]. This huge phenotypical variation makes it challenging to define and predict disease severity and outcomes [9]. Additionally, in SCD, numerous biomarkers were identified to be involved in the disease pathophysiology [10], but their clinical utility has not been achieved yet, as several factors contribute to inflammation, oxidative stress, adhesion, and coagulation [11,12]. Nevertheless, no genetic marker for VOC has been identified yet.

This study aimed to determine the differentially expressed genes in SCD patients in steady-state and in VOC compared to healthy controls using microarray. As well as investigating the effect of VOC on SCD gene expression analysis to detect genetic markers associated with the VOC. The control group consisted of eight healthy Bahraini volunteers with confirmed Hemoglobin AA genotype, as determined by High-Performance Liquid Chromatography (HPLC). These participants had no history of sickle cell disease or related hemoglobinopathies, ensuring a baseline for comparison with SCD patients in steady-state and VOC. Such a study may enhance our knowledge of disease pathophysiology and may aid in the development of targeted therapy to treat and prevent VOC.

## 2. Results

### 2.1. Characteristics of Participants

The study enrolled twenty sickle cell disease patients, of whom ten patients had VOC. The summary of the patient’s characteristics is shown in Table 1. The statistical analysis showed no significant differences between both of the two groups in baseline characteristics (*p*-value > 0.05).

### 2.2. Determination of the Differentially-Expressed Genes

The analysis of microarrays expression data identified 2073 dysregulated genes in which 736 genes were up-regulated (*p* < 0.05) with a fold change of >2, and 1337 genes were down-regulated (*p* < 0.05) with a fold change of <−2 in SCD patients in steady-state compared to healthy controls, as seen in (Figure 1A). Whereas in SCD patients in VOC compared to healthy controls, 3363 genes were differentially regulated, including 1080 genes that were up-regulated (*p* < 0.05) with a fold change of >2, and 2283 genes were down-regulated (*p* < 0.05) with a fold change of <−2 (Figure 1B). In addition, 1078 genes were differentially expressed, including 410 up-regulated genes and 668 down-regulated genes in SCD patients in VOC compared to steady-state (*p* < 0.05) with a fold change of >2 and <−2, respectively, as seen in Figure 1C.

Furthermore, the assessment of down-regulated genes revealed 47 genes to be down-regulated in SCD patients in steady-state compared to healthy controls at a *p*-value of <0.001 and a fold change of >4, while in SCD patients in VOC compared to healthy controls, 255 genes were down-regulated (Table 2). Additionally, 79 genes were down-regulated in SCD patients in VOC compared to steady-state

### 2.3. Potential Genetic Marker for Vaso-Occlusive Crisis

To identify potential genetic markers for VOC, a GO term enrichment analysis was conducted to reveal key biological processes affected during this condition. The analysis showed a significant up-regulation of genes associated with processes like cellular localization, signal transduction regulation, and ubiquitin ligase complex involvement (Figure 2). These terms suggest an increase in cellular activity and regulatory functions, likely indicative of heightened cellular demands and stress responses associated with the inflammatory and hypoxic environment during VOC. This increased activity may be central to the mechanisms of cell damage and adaptation underlying the Vaso-occlusive state.

In contrast, down-regulated genes were notably associated with terms related to immune system processes, MHC protein complex binding, and T cell receptor binding (Figure 2). These findings suggest a suppression of immune response pathways and a reduction in cellular activities essential for maintaining immune cell function and cellular homeostasis. Such down-regulation of immune-related and hematopoietic pathways could impact immune signaling and cellular stability, possibly contributing to impaired cellular responses in SCD patients experiencing VOC.

In focusing further on genes with significant deregulation, particularly those implicated in immune and hematopoietic regulation, we identified a subset of 79 down-regulated genes. To refine our search for a robust VOC marker, we selected those genes that were consistently down-regulated with a four-fold change in SCD patients during VOC compared to steady-state and healthy controls (*p*-value < 0.001) but not more than two-fold up-regulated in steady-state patients compared to healthy controls (*p*-value < 0.05). This filtering left 26 genes (Table 3) with a high likelihood of true down-regulation during VOC.

Among these, RUNX3 (RUNX Family Transcription Factor 3) emerged as a promising candidate for further analysis. RUNX3 showed a four-fold down-regulation in SCD patients during VOC compared to steady-state levels (Figure 3).

### 2.4. Validation of RUNX3 Through qRT-PCR and ELISA

To validate the microarray data with the performed qRT-PCR and ELISA assays to assess the gene expression and protein levels of RUNX3 in the same samples. The analysis of RUNX3 gene expression by qRT-PCR showed statistically significant down-regulation in SCD patients in VOC compared to SCD patients in steady-state with three-fold changes (*p* = 5.517× 10^−6^), as seen in Figure 4A,C. Moreover, the measurement of RUNX3 protein concentration using ELISA showed a significant reduction in the concentration in SCD patients in VOC (11.13 pg/mL) compared to SCD patients in steady-state (457.93 pg./mL) (*p* = 5.957 × 10^−11^) Figure 4B).

## 3. Discussion

In SCD, gene expression meta-analysis studies conducted on the West African population identified several biological pathways to be associated with SCD through enrichment analysis. The innate immunity pathway was recognized among the major pathways [13,14]. The IL7R (Interleukin 7 Receptor) and TRAT1 (T Cell Receptor Associated Transmembrane Adaptor 1) genes were among the top down-regulated in the previous meta-analysis studies, as well as in our study. The IL7R protein is involved in the adaptive immune system and plays a crucial part in V(D)J recombination during lymphocyte development [15]. Also, it has a relation with other pathways such as hematopoietic cell lineage and extracellular signal-regulated kinases [16,17]. Whereas the TRAT1 gene is a protein-coding gene that plays a role in the adaptive immune system as it stabilizes the T-cell antigen receptor (TCR)-CD3 complex at the surface of T-cells and is related to pathways in downstream signaling [18]. Moreover, the CD3E (CD3e Molecule) gene, which coded for CD3-epsilon polypeptide, was remarkably down-regulated in the steady-state and VOC group. This protein is involved in the activation of downstream signaling pathways, it has an important role in T-cell development and initiation of TCR-CD3 complex assembly [19].

Furthermore, our study was the first to enroll Arabs with SCD in whole-blood gene expression analysis. In our study, RUNX3 (RUNX Family Transcription Factor 3) was significantly down-regulated in SCD patients in VOC compared to SCD patients in steady-state, and this was further confirmed by qRT-PCR and ELISA. These results suggest that the RUNX3 gene may serve as a potential genetic marker for the development of VOC in SDC patients. Additionally, RUNX3 was not identified in previous studies characterizing the gene expression in SCD patients, and no study assessed its role at the transcriptional level in SCD patients.

RUNX3 is a member of the runt domain-containing family of transcription factors located on chromosome 1p36.11 and formed of 65,647 bases [20]. It functions in activating or suppressing transcription through binding of its heterodimer protein form to the core DNA sequence 5′-TGTGGT-3′ in some enhancers and promoters after forming a complex with beta subunit forms (heterodimeric complex core-binding factor with Core-Binding Factor Subunit Beta (CBFB)), as well as interacting with other transcription factors [21]. Also, it works as a tumor suppressor as it is found to be silenced or deleted in cancer [22,23]. Additionally, it may have a role in controlling cellular proliferation and differentiation [22].

Several studies assessed the role of RUNX3 in malignancies such as gastric, colon, and lung cancers [24,25]. Besides, dysfunction of RUNX3 resulted in defects of transcriptional regulation and DNA repair, leading to bone marrow failure, which may progress to leukemia [26]. Moreover, Yanyan and co-workers reviewed data from animal and human studies which related RUNX3 to the pathogenesis of bronchial asthma part of it due to its involvement in the regulation of Th1/Th2 balance and the other part associated with its effect in modifying the several immune cell differentiations such as innate lymphoid cells, Treg cells, and dendritic cells [27]. In some reports, it was suggested that oxidative stress induced phosphorylation of RUNX3, leading to its cytoplasmic localization and subsequent inactivation, mediating carcinogenesis along with hypersensitivity [28,29]. The effect of RUNX3 on Th1/Th2 balance was through its involvement in class I Major Histocompatibility Complex assortment of T cells, particularly CD8+ during their development, and it was found that silencing of RUNX3 expression may inhibit Th1 cytokine release while facilitating Th2 cytokine secretion [30]. Additionally, it has a critical function in the differentiation of innate lymphoid cells by altering CD8+ T cells, Th1, and spleen natural killer cells differentiation [31], which in turn can predispose to the development of inflammatory disease [32].

Moreover, RUNX3 was found to be protective against acute lung injury in rats with severe acute pancreatitis as up-regulation of RUNX3 resulted in increasing polymorphonuclear neutrophil apoptosis and inhibition of Janus kinase 2/signal transducer and activator of transcription 3 (JAK2/STAT3) phosphorylation which in turn decreases the progression of inflammatory response and subsequently organ damage [33].

Furthermore, Das et al. investigated the role of microRNA in modulating the HbF induction pathway by comparing the gene expression between individuals with high HbF levels, including subjects with hereditary persistence of fetal hemoglobin and β-thalassemia minor, with those having normal HbF. The study identified 931 differentially regulated genes and 19 differentially expressed miRNAs. In which two miRNAs were inversely correlated with RUNX3 mRNA: high expression of miRNA-301b and miRNA-582-5p resulted in decreased expression of RUNX3 [34]. Interestingly, the up-regulation of miRNA-301b was suggested to be triggered by hemolysis and hypoxia [35,36]. Add together, decreased expression of RUNX3 disrupts the inflammatory balance, predisposing to tissue inflammation and injury, which forms the basis of VOC pathophysiology.

### 3.1. Molecular Mechanisms of RUNX3 in VOC Pathophysiology

(1)Immune System Regulation: Based on our GO term enrichment analysis (Figure 2), RUNX3 down-regulation significantly impacts immune system processes, particularly through: (a) T-Cell Development and Function: RUNX3′s role in CD8+ T cell development aligns with our finding of down-regulated T cell receptor binding pathways.This is supported by previous studies showing RUNX3′s critical function in T-cell lineage specification [30]. The significant down-regulation of IL7R (12.76-fold) and CD3E (11.28-fold) in our study further supports this immune dysregulation pathway(2)Inflammatory Response: Our findings showed significant alterations in inflammatory pathways, specifically: (a) Cytokine Regulation: RUNX3 down-regulation correlates with altered inflammatory mediator expression. This is evidenced by the substantial down-regulation of CCL5 (9.24-fold) in our VOC samples. Previous studies have shown RUNX3′s protective role against inflammatory response through regulation of the JAK2/STAT3 pathway [33].(3)Cellular Response Mechanisms: The GO term enrichment analysis revealed: (a) Signal Transduction: Up-regulation of cellular localization and signal transduction regulation pathways. These changes suggest adaptive responses to the inflammatory and hypoxic environment during VOC. The diagram below illustrates the implications of RUNX3 gene downregulation on various molecular pathways and biological processes, as suggested by our g:Profiler results.



### 3.2. Integration with VOC Pathophysiology

The molecular changes we observed suggest that RUNX3 down-regulation contributes to VOC through:(a)Immune System Disruption: Impaired T-cell development and function, evidenced by down-regulation of key T-cell markers (IL7R, CD3E). Altered MHC protein complex binding, affecting immune response regulation(b)Inflammatory Cascade: Disrupted cytokine production and signaling. Altered inflammatory mediator expression (CCL5 down-regulation).(c)Cellular Stress Response: Enhanced cellular localization and signal transduction

Increased ubiquitin ligase complex activity, suggesting altered protein homeostasis

These molecular mechanisms provide a comprehensive framework for understanding how RUNX3 down-regulation contributes to VOC pathophysiology and supports its potential as a biomarker for VOC in SCD patients.

The GO terms enrichment analysis shown in Figure 2 from the manuscript is used to create a more precise molecular mechanism discussion:

### 3.3. Molecular Mechanisms Based on GO Term Enrichment Analysis

Up-regulated Pathways During VOC: From our GO term analysis (Figure 2), three major up-regulated processes were identified: (a) Cellular Localization: Enhanced protein localization to the cell periphery and increased membrane protein localization. These changes suggest active cellular reorganization during VOC. (b) Signal Transduction Regulation: Increased regulation of signal transduction and enhanced cellular response to stimuli. These pathways indicate adaptive responses to VOC conditions. (c) Ubiquitin Ligase Complex: Up-regulation of ubiquitin protein ligase binding and enhanced protein modification processes. This suggests increased protein turnover and cellular stress response

A detailed suggested diagram for visualization of these GO terms:



### 3.4. Timeline Limitations

This cross-sectional study, conducted in September 2019, collected samples within a specific window (e.g., within 48 h for VOC patients), limiting the ability to capture longitudinal gene expression dynamics. A single time-point analysis may not fully reflect the variability of vaso-occlusive crises (VOC), which are episodic and influenced by environmental, physiological, or therapeutic factors. Consequently, the study could not assess whether RUNX3 down-regulation is consistent across recurrent VOCs or varies with disease progression. Longitudinal studies tracking patients over months or years could elucidate the stability of RUNX3 as a biomarker and its correlation with VOC frequency, severity, or response to treatments like hydroxyurea, which was not used in this cohort.

### 3.5. Functional Confirmation Needs

While the study identifies RUNX3 as a potential VOC biomarker, functional confirmation is needed to establish causality and mechanistic roles. In vitro studies manipulating RUNX3 expression (e.g., via CRISPR/Cas9 in T-cells or hematopoietic cells) could assess its impact on immune cell differentiation, cytokine production (e.g., Th1/Th2 balance), and inflammation under VOC-like conditions (e.g., hypoxia). In vivo studies using SCD animal models (e.g., transgenic sickle mice) could evaluate RUNX3′s role in vaso-occlusion and tissue inflammation. Protein interaction studies (e.g., co-immunoprecipitation or ChIP-seq) could map RUNX3′s transcriptional targets, clarifying its regulation of immune and inflammatory pathways. Additionally, correlating RUNX3 levels with clinical outcomes (e.g., VOC frequency or organ damage) in larger cohorts would validate its predictive value. These studies would confirm whether RUNX3 down-regulation drives VOC or is a secondary effect, elucidate mechanistic pathways, validate biomarker reliability across diverse SCD populations, and explore therapeutic potential (e.g., via gene therapy). Addressing these gaps would enhance the study’s rigor and clinical relevance, responding to concerns about the preliminary nature of the findings due to the small sample size (*n* = 28) and lack of mechanistic data.

In conclusion, analysis of the transcriptional changes in SCD during steady-state and VOC resulted in the detection of genes for the first time to be associated with SCD and were significantly differentially expressed. Amongst these genes, RUNX3 plays a role in immune cell differentiation and inflammation. It may serve as a potential genetic biomarker, and further validation in a larger sample size is recommended. Additionally, studying the RUNX3 pathway in relation to disease pathogenesis may aid in the discovery of novel therapeutic targets and enable the development of personalized medicine in managing SCD patients. Nonetheless, the study is subject to some limitations, including the small sample size, the timeline, and the need for a functional confirmation study.

## 4. Materials and Methods

### 4.1. Study Population

Twenty Bahraini patients with SCD (10 at steady-state and 10 with VOC) and eight healthy participants from Salmaniya Medical Complex were recruited in this cross-sectional study in September 2019. Approval from the Research and Ethics Committee of the Arabian Gulf University and the Research Technical Support Team of the Ministry of Health, Kingdom of Bahrain, was obtained. Written informed consents were obtained from each study participant. All experiments and methods were performed in accordance with relevant guidelines and regulations.

Healthy volunteers were confirmed to have the Hemoglobin AA genotype by High-Performance Liquid Chromatography (HPLC), while all patients with SCD were confirmed to have the HbSS genotype. SCD patients were divided into two groups of ten participants each: SCD patients in a steady-state defined as participants without any history of VOC that required neither evaluation in an emergency department nor hospital admission 12 weeks prior to the study enrollment; and SCD patients during VOC-defined as participants with a history of acute, severe pain at the time of enrollment (self-rated score of ≥7 out of 10 on a Numerical Rating Scale (NRS)) [37,38]. Not all SCD patients were under treatment with hydroxyurea.

### 4.2. Sample Collection

For all subjects, blood samples were collected in two separate tubes. For the VOC group, the samples were collected within the first 48 h of the crisis. First, 5 milliliters of venous blood were collected in a serum-separating tube and kept for 30 min at room temperature for clot formation, and then centrifuged at 3500 rpm for 15 min. The separated serum was stored at −80 °C until the analysis. In the second container, 2.5 milliliters of venous blood were collected in PAXgene^®^ Blood RNA Tube (PreAnalytiX GmbH, Hombrechtikon, Switzerland) for immediate stabilization of intracellular RNA and was kept for a minimum of 2 h at room temperature to allow for complete lysis of blood cells and then stored at 4 °C until the analysis that was carried out within 3 days.

### 4.3. RNA Extraction and Gene Expression Analysis

RNA was extracted from whole blood samples using a PAXgene^®^ Blood RNA kit (PreAnalytiX GmbH, Hombrechtikon, Switzerland) following the manufacturer’s instructions. The quantity and purity of RNA samples were determined using the NanoDrop 1000 Spectrophotometer (Thermo Fisher Scientific, Inc., Waltham, MA, USA), and the acceptable RNA purity of A260/A280 was 1.8–2.2. The RNA integrity was assessed using 1.2% agarose gel electrophoresis. All RNA samples were stored at −80 °C until further analysis.

The assessment of gene expression was carried out using Affymetrix ClariomTM S Assays for human and GeneChip™ WT PLUS Hybridization, Wash and Stain Kit (Applied Biosystems™, Foster, CA, USA) according to the manufacturer’s protocol. In brief, the reverse transcription of 100 ng of total RNA of each sample was converted to double-stranded cDNA using the T7 promoter sequence primer. Followed by the synthesis and amplification of cRNA by an in vitro transcription of the second-stranded cDNA using T7 RNA polymerase. Then, through reverse transcription of cRNA, the second cycle of single-stranded cDNA was synthesized, which contains dUTP. After the hydrolysis of the RNA, 5.5 μg of purified single-stranded cDNA was fragmented using uracil-DNA glycosylase and apurinic/apyrimidinic endonuclease 1. Next, by terminal deoxynucleotidyl transferase (TdT), the fragmented cDNA was labeled with DNA Labeling Reagent, which binds to biotin. The fragmented and biotin-labeled single-stranded cDNA samples were hybridized to GeneChip™ WT PLUS for sixteen hours in the Affymetrix GeneChip^®^ Hybridization Oven 645 (Affymetrix, Santa Clara, CA, USA). This was followed by washing and staining using the Affymetrix GeneChip^®^ Fluidics Station 450 (Affymetrix, Santa Clara, CA, USA) and the Affymetrix^®^ GeneChip^®^ Command Console™ (AGCC) software version 4.0 (Affymetrix, Santa Clara, CA, USA). Finally, the arrays were scanned using the Affymetrix GeneChip^®^ Scanner 3000 7G (Affymetrix, Santa Clara, CA, USA).

### 4.4. Quantitative Real-Time Polymerase Chain Reaction

Real-time polymerase chain reaction (PCR) was performed to measure the expression of the RUNX Family Transcription Factor 3 (RUNX3) gene and normalized to GAPDH as a housekeeping gene. The reaction mixture for the SYBR Green assay contained 2× SYBR™ Select Master Mix (Applied Biosystems™, Foster, CA, USA), 10 pmol of each forward and reverse primer (Metabion International AG, Planegg, Germany), and 50 ng of cDNA.

The sequences of the primers for RUNX3 and GAPDH were as follows: RUNX3 forward primer, 5′-GGCAATGACGAGAACTACTCCG-3′, and RUNX3 reverse primer 5′-GATGGTCAGGGTGAAACTCTTCC-3′; GAPDHforward primer 5′-TCCCTGAGCTGAACGGGAAG-3′, and GAPDH reverse primer 5′-GGAGTGGGTGTCGCTGT-3′.

The reaction was carried out in 20 µL capillaries and incubated in Light Cycler^®^ 2.0 (Roche, Basel, Switzerland). The LightCycler run protocol is as follows: Denaturation program at 95 °C for 10 min, amplification, and quantification program repeated 45 times at 95 °C for 10 s, 60 °C for 30 s, and 72 °C for 30 sec, and finally a cooling step at 40 °C for 30 s. The accumulation of PCR products during each cycle was determined by observing the rise in fluorescence of DNA-binding SYBR Green. Afterward, the crossing point of each sample was detected and normalized to the expression of the housekeeping gene. Then, the fold change of expression was calculated using the 2^−ΔΔCt^ method.

### 4.5. Enzyme-Linked Immunosorbent Assay

The protein produced by the RUNX3 gene was measured by Enzyme-Linked Immunosorbent Assay (ELISA) (MyBioSource, San Diego, CA, USA, Cat. No: MBS9332568) according to the manufacturer’s instructions. All standards and samples were assigned duplicates to the plate, and the assay was run by adding 50 µL per well of diluted standard to the standard well. Then, 40 μL of Sample Diluent was pipetted to each sample well and combined with 10 μL of sample. After incubation and washing, 50 µL of HRP-conjugated detection antibody was added (HRP-conjugated Human RUNX3 detection antibody), incubated, and washed. After that, 50 µL of each chromogen solution A and B were pipetted into each well, incubated, and washed. Then, 50 µL of Stop Solution was added for reaction termination when the desired blue color intensity was attenuated. Immediately, the optical density (OD) at 450 nm was measured for each well using Synergy™ HTX Multi-Mode Microplate Reader (BioTek^®^) (BioTek Instruments, Inc., Winooski, VT, USA) and analyzed by Gen5 2.07.17.

### 4.6. Statistical Analysis

The demographic data was analyzed for their differences in SCD patients in steady-state and during VOC. The values of continuous data were analyzed by Student’s two-sided unpaired *t*-test and presented as mean ± standard deviation (SD). The categorical variables were presented in numbers (percentages) and were analyzed using Fisher’s exact probability test. A *p*-value of <0.05 was considered significant.

The Transcriptome Analysis Console (TAC) software version 4.0.0.25 by Thermo Fisher Scientific was used to define the differential expression profile within the different groups, perform statistical analysis, and provide a list of differentially expressed genes. Genes with a fold change of >2 or <−2 and with a *t*-test or ANOVA *p*-value of <0.05 were considered significantly altered between the conditions of each group.

The Light Cycler Software version 4.1.1.21 was used for the analysis of qRT-PCR results by identifying the crossing points for the target and the reference gene in each sample. The average of crossing points for each target gene was calculated relative to the housekeeping gene GAPDH in all the groups using the relative Mono-Color Relative Quantification assay. Then the fold change was calculated using the delta Ct (2^−ΔΔCt^) method.

### 4.7. Gene Ontology (GO) Term Enrichment Analysis

The GO term enrichment analysis was performed to identify differentially deregulated GO terms between SCD patients in steady-state and during VOC using the g: Profiler (version: e111_eg58_p18_f463989d). g: Profiler enables comprehensive analysis of Gene Ontology (GO) terms, covering the biological process (BP), cellular component (CC), and molecular function (MF) domains to investigate pathway enrichment across differentially expressed genes. Input gene lists were based on significant differential expression results, with criteria set for adjusted *p*-values (threshold: FDR < 0.05) to ensure a high-confidence identification of GO terms [39].

The enrichment results were visualized as bar plots, illustrating the distribution and -log10-transformed adjusted *p*-values for up-regulated and down-regulated GO terms, thereby highlighting terms with significant deregulation between the VOC and SCD samples.

The datasets generated during and/or analyzed during the current study are available from the corresponding author on reasonable request.

## Figures and Tables

**Figure 1 ijms-26-06338-f001:**
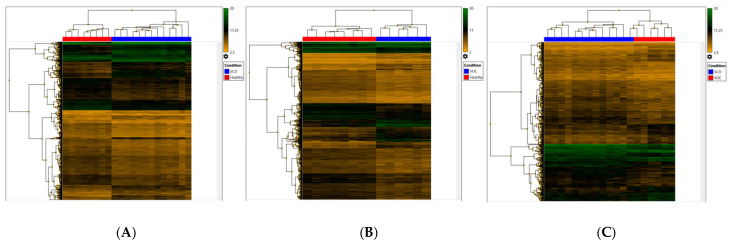
Hierarchical Clustering of the differentially regulated genes at a *p*-value of <0.05 and a fold change of <−2 or >2: (**A**) 2073 differentially regulated genes in SCD patients in steady-state compared to healthy controls; (**B**) 3363 differentially regulated genes in SCD patients in VOC compared to healthy controls; (**C**) 1078 differentially regulated genes in SCD patients in VOC compared to SCD patients in steady-state.

**Figure 2 ijms-26-06338-f002:**
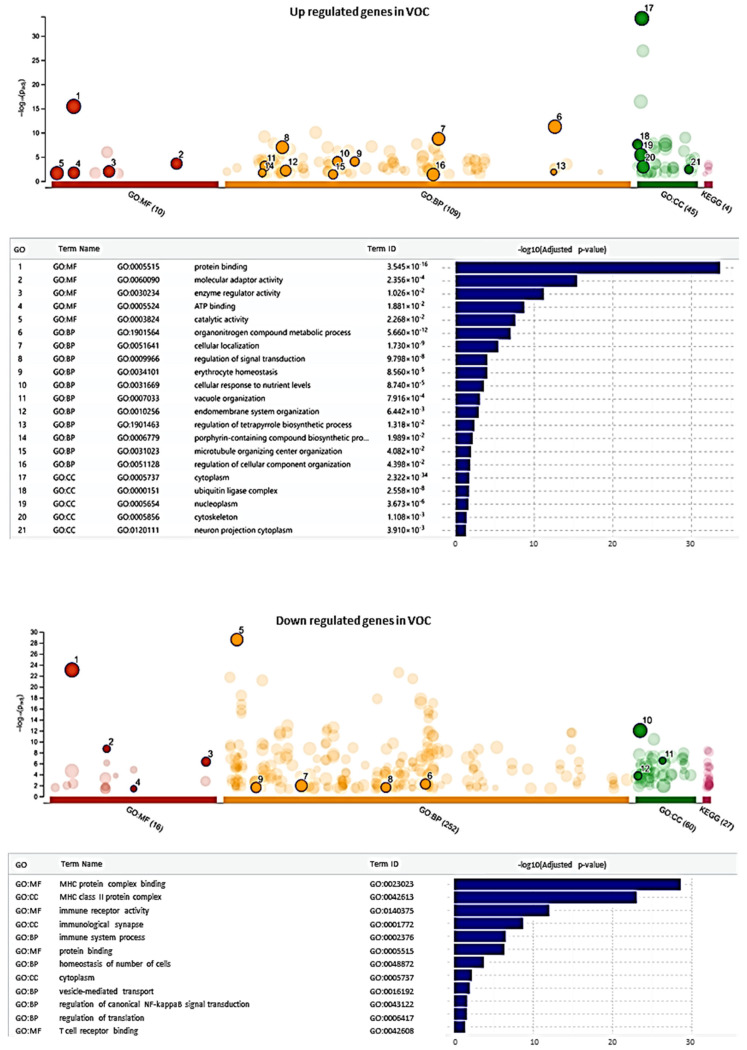
Implications of RUNX3 Gene Down and Up-regulation on Molecular Pathways and Biological Processes. This diagram illustrates the effects of RUNX3 gene downregulation on various molecular pathways and biological processes, as suggested by our g:Profiler results. Key pathways and processes affected include the activation of the immune system (red), inhibition of the inflammatory response (green), and moderate effects on signal transduction pathways (yellow). Complex interactions involving T cell receptor binding are indicated in blue, while the MHC protein complex shows moderate activation (orange). The development of CD8+ T cells is activated (red), and CCL5 expression is inhibited (green). The JAK2/STAT3 pathway exhibits moderate effects (yellow), and cytokine production is activated (red). Cellular localization remains neutral (blue), and complex interactions involving the ubiquitin ligase complex are also indicated in blue. The stress response shows moderate activation (orange). The colors used in the diagram (red, green, blue, yellow, orange) help to visually distinguish the different states and interactions of these pathways and processes.

**Figure 3 ijms-26-06338-f003:**
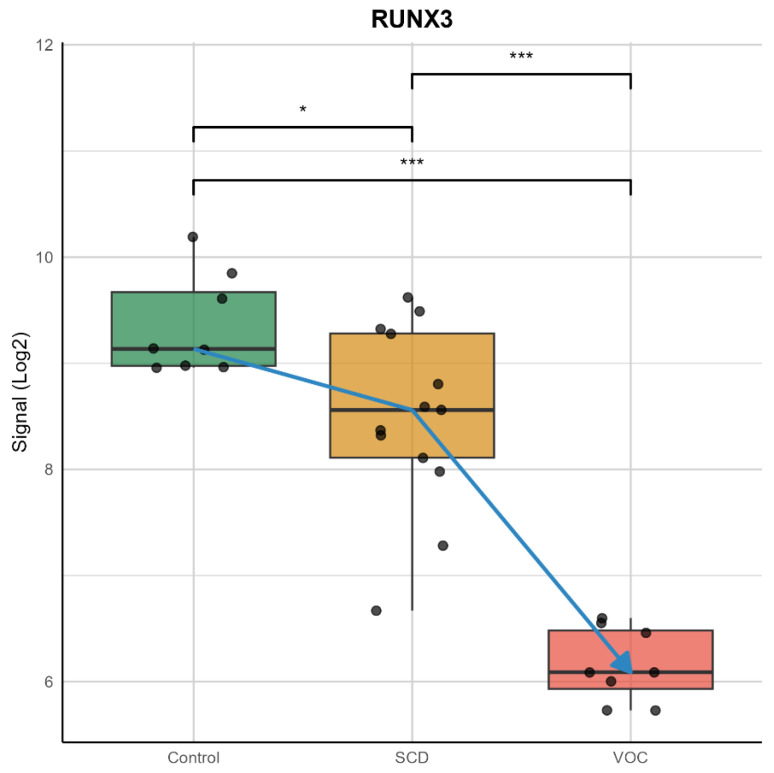
Signal Intensity of RUNX3 Gene in Control and Experimental Conditions. The plot shows the signal intensity of the RUNX3 gene across various samples conditions. The control condition (green) exhibits a relatively stable signal intensity. In contrast, the experimental conditions (orange box for SCD and red box for VOC) show more variability and lower signal intensity. The black dots represent individual reading points for each patient in the respective groups, providing a detailed view of the data. The blue arrow highlights a specific decrease in signal intensity, among different groups. Significance of *p*-values: Control vs. VOC: The *p*-value is significant at the 0.001 level (indicated by ***). SCD vs. VOC: The *p*-value is significant at the 0.05 level (indicated by *). These *p*-values indicate that there are statistically significant decreases in the signal intensity of the RUNX3 gene from the control to the SCD conditions, from the control to the VOC conditions, and from the SCD to the VOC conditions. The highest significance is observed between the control and VOC conditions, followed by the control and SCD conditions, and the SCD and VOC conditions.

**Figure 4 ijms-26-06338-f004:**
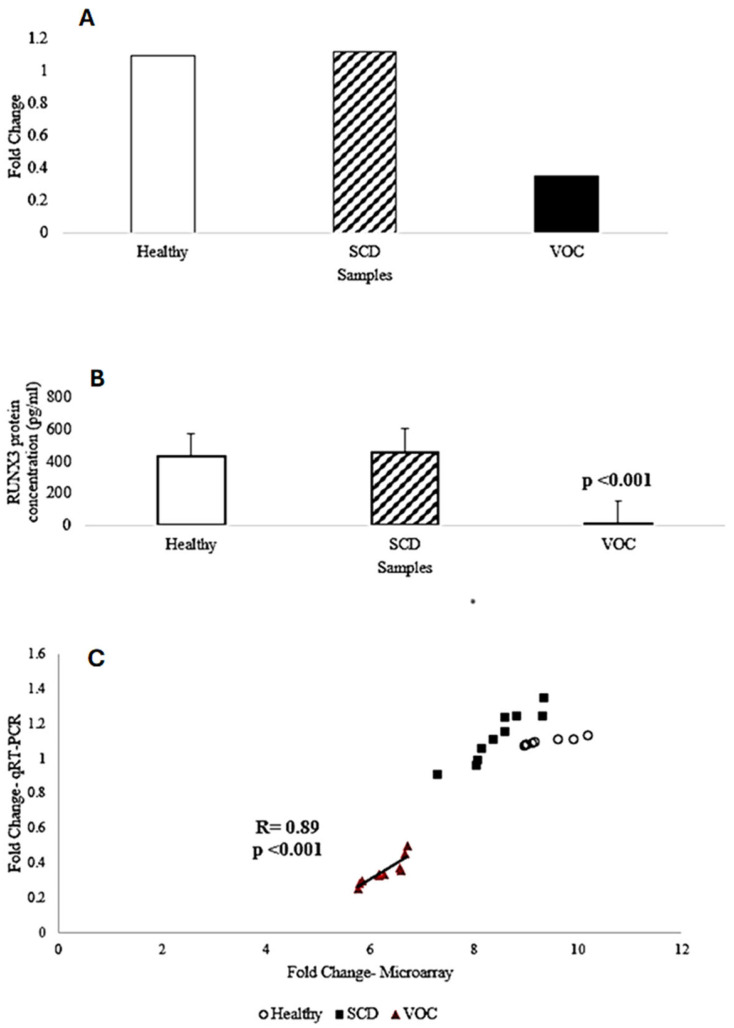
RUNX3 gene expression level and protein concentration: (**A**) Average fold change of RUNX3 gene through real-time polymerase chain reaction (qRT-PCR) showing three-fold down-regulation in SCD patients in VOC compared to SCD patients in steady-state at a *p*-value of 5.517 × 10^−6^; (**B**) The average RUNX3 protein concentration was reduced in SCD patients in VOC compared to SCD patients in steady-state at a *p*-value of 5.957 × 10^−11^; (**C**) Correlation of fold change of RUNX3 gene expression measured by microarray and qRT-PCR. The x-axis represents log2-fold change determined by microarrays; the y-axis represents log2-fold change determined by qRT-PCR.

**Table 1 ijms-26-06338-t001:** Baseline characteristics of the study participants with SCD (*n* = 20).

Parameters		Steady-State	VOC	*p*-Values
Gender [*n* (%)]	Male	9 (90)	9 (90)	1
	Female	1 (10)	1 (10)	
Age Mean in years ± SD		33 ± 10.82	34.9 ± 9.3	0.68
No. of VOC per year ± SD		9.7 ± 6.16	8.3 ± 5.38	0.59
No. of Hospital admissions per year Mean ± SD		3.3 ± 1.83	3.7 ± 1.64	0.61
White Blood Cell counts Mean in × 10^9^/L ± SD		5.4 ± 2.95	6.05 ± 3.91	0.68
Red Blood Cell counts Mean in × 10^12^/L ± SD		4.89 ± 0.94	4.07 ± 0.95	0.07
Hemoglobin Mean in g/dL ± SD		11.17 ± 1.14	10.48 ± 1.51	0.27
Platelet Mean in × 10^9^/L ± SD		309.19 ± 205.84	201.2 ± 118.04	0.17
Hemoglobin F Mean in % ± SD		13.88 ± 8.3	18.26 ± 6.02	0.2
Hemoglobin S Mean in % ± SD		79.81± 7.97	76.25± 5.41	0.26

**Table 2 ijms-26-06338-t002:** Top ten down-regulated genes at *p*-values of <0.001 and a fold change of >4.

ID	Gene Symbol	Chromosome	Group	*p*-Values	Fold Change
SCD Patients in Steady-State Compared to Healthy Controls					
TC1100010092.hg.1	*EIF4G2*; *SNORD97*	chr11	Multiple_Complex	1.31 × 10^−10^	−16.33
TC0200007835.hg.1	*ACTR2*	chr2	Multiple_Complex	3.00 × 10^−11^	−8.22
TC1500009865.hg.1	*ANP32A*	chr15	Multiple_Complex	1.79 × 10^−9^	−6.67
TC1700007383.hg.1	*RPL23A*; *SNORD4B*; *SNORD42B*; *SNORD42A*	chr17	Multiple_Complex	1.93 × 10^−11^	−6.6
TSUnmapped00000264.hg.1	*RPL7A*		Coding	2.88 × 10^−12^	−6.47
TC0700010538.hg.1	HNRNPA2B1	chr7	Multiple_Complex	3.42 × 10^−12^	−6.07
TC0600007378.hg.1	HIST1H4J	chr6	Coding	1.34 × 10^−8^	−5.81
TC1500008312.hg.1	IQGAP1	chr15	Multiple_Complex	1.18 × 10^−5^	−5.64
TC0800010667.hg.1	PDE7A	chr8	Multiple_Complex	2.26 × 10^−8^	−5.64
TC0600011227.hg.1	HIST1H4K	chr6	Coding	1.66 × 19^−8^	−5.6
SCD Patients in VOC Compared to Healthy Controls					
TC0500007138.hg.1	IL7R	chr5	Multiple_Complex	2.75 × 10^−10^	−12.76
TC1100009200.hg.1	CD3E	chr11	Multiple_Complex	2.61 × 10^−12^	−11.28
TC1700012052.hg.1	ACTG1	chr17	Multiple_Complex	2.01 × 10^−7^	−10.4
TC1200007758.hg.1	HNRNPA1	chr12	Multiple_Complex	6.70 × 10^−10^	−9.75
TC0600011173.hg.1	GUSBP2	chr6	Multiple_Complex	2.47 × 10^−7^	−9.28
TC1700010447.hg.1	CCL5	chr17	Coding	4.51 × 10^−8^	−9.24
TC0500013430.hg.1	GNB2L1; SNORD95; SNORD96A	chr5	Multiple_Complex	6.41 × 10^−9^	−8.96
TC0100007676.hg.1	LCK	chr1	Multiple_Complex	8.11 × 10^−10^	−8.92
TC0400011548.hg.1	LEF1	chr4	Multiple_Complex	7.22 × 10^−9^	−8.88
TC0600011517.hg.1	HLA-DPA1	chr6	Multiple_Complex	6.94 × 10^−7^	−8.79

**Table 3 ijms-26-06338-t003:** Differentially regulated genes in SCD patients in VOC compared to SCD patients in steady-state.

ID	Gene Symbol	Chromo-Some	*p*-Values	Fold Change		
				VOC vs. Steady-State	VOC vs. Healthy	Steady-State vs Healthy
TC0100017110.hg.1	FCMR	chr1	2.75 × 10^−7^	−10.98	−9.24	−1.98
TC0200008268.hg.1	GNLY	chr2	1.76 × 10^−5^	−7.37	−8.92	−1.48
TC1200010850.hg.1	TESPA1	chr12	2.22 × 10^−5^	−6.3	−8.72	−1.48
TC1700010447.hg.1	CCL5	chr17	7.39 × 10^−7^	−6.24	−7.92	−1.46
TC0600007657.hg.1	HLA-DQA1	chr6	3.97 × 10^−5^	−6.1	−6.5	−1.41
TC1900011774.hg.1	EMP3	chr19	2.17 × 10^−7^	−5.99	−6	−1.34
TC0500012470.hg.1	CD74	chr5	1.06 × 10^−6^	−5.89	−5.9	−1.32
TC1200006738.hg.1	KLRG1	chr12	0.0001	−5.72	−5.83	−1.19
TC1200012571.hg.1	ITFG2	chr12	8.44 × 10^−9^	−5.52	−5.82	−1.15
TC2200008641.hg.1	RAC2	chr22	9.48 × 10^−7^	−5.43	−5.8	−1.07
TC0200011075.hg.1	PTMA	chr2	3.16 × 10^−8^	−5.41	−5.76	−1.07
TC1000011904.hg.1	ABLIM1	chr10	5.35 × 10^−7^	−5.22	−5.69	−1.06
TC1600006888.hg.1	CIITA	chr16	1.76 × 10^−7^	−5.21	−5.69	1
TC1200012801.hg.1	CS	chr12	2.63 × 10^−6^	−5.19	−5.34	1.01
TC1200012583.hg.1	CD27	chr12	2.89 × 10^−5^	−5.14	−4.99	1.03
TC1100013190.hg.1	CFL1	chr11	6.80 × 10^−7^	−4.97	−4.59	1.05
TC0600007650.hg.1	HLA-DRA	chr6	7.56 × 10^−6^	−4.96	−4.53	1.1
TC1900007839.hg.1	FXYD5	chr19	2.52 × 10^−8^	−4.95	−4.53	1.12
TC1200010616.hg.1	TUBA1B	chr12	1.20 × 10^−7^	−4.8	−4.36	1.14
TC0100007676.hg.1	LCK	chr1	6.31 × 10^−7^	−4.52	−4.31	1.14
TC0300006993.hg.1	CRTAP	chr3	2.83 × 10^−6^	−4.4	−4.27	1.15
TC1100007790.hg.1	CD5	chr11	8.94 × 10^−8^	−4.39	−4.18	1.16
TC1400006656.hg.1	OXA1L	chr14	1.78 × 10^−8^	−4.36	−4.16	1.18
TC0100013339.hg.1	RUNX3	chr1	2.33 × 10^−8^	−4.12	−4.08	1.22
TC0100018246.hg.1	LRRC8C	chr1	1.94 × 10^−6^	−4.09	−4.04	1.3
TC0800009819.hg.1	DOK2	chr8	3.24 × 10^−6^	−4.07	−4.03	1.32

## Data Availability

Data will be available upon request.

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
