# Peer review of "The Whole Blood Transcriptomic Analysis in Sickle Cell Disease Reveals RUNX3 as a Potential Marker for Vaso-Occlusive Crises"

_ijms, 2025, doi:10.3390/ijms26136338_

Round 1
Reviewer 1 Report
Comments and Suggestions for Authors
The authors have performed a whole blood transcriptome analysis of patients with sickle cell disease, before (n=10) and during (n=10) vaso-occlusive crisis (VOC). A fatal flaw is that these two groups are composed of different patients. Other comments are that the technology used (Affymetrix microarrays) is vastly outdated; the healthy control population is not described in any detail (number of participants, male/female, standard blood parameters); the analysis of RUNX3 analysis by ELISA is not convincing and has not been described in sufficient detail. Which antibody was used for detection of RUNX3? Why were the results not validated by western blot analysis which would provide important additional information such as the size of the protein and potentially the presence of isoforms?
Minor comment: In places the manuscript requires some copy editing and removal of instructions for the authors.
Author Response
Reviewer 1
Thank you very much for your time in reviewing our work. Your comments and advice are valuable and helpful in improving our paper.
The authors have performed a whole blood transcriptome analysis of patients with sickle cell disease, before (n=10) and during (n=10) vaso-occlusive crisis (VOC). A fatal flaw is that these two groups are composed of different patients.
Our study aimed to identify the differentially regulated gene during VOC and we use the steady-state as a control. We didn’t aim to study a single gene, so we followed it up in the same group. Additionally, the definition of study-state (when there is no recent drop in the hemoglobin level and there is absence of infection, pain, acute complicating factors or acute clinical symptoms or crisis for at least three months established by a careful history and complete physical examination) based on Bookchin and Lew (https://doi.org/10.1016/S0889-8588(05)70397-X) where many studies done on SCD patients in steady-state followed the same definition. As shown in Table 1 the mean VOC/year is 8 so it is hard to say that the patient will not get another attack before completing the 3 months period. The study was done for a master program and had a limited timeline. However, for further large-scale study where we will work on modulating the RUNX3 gene expression defiantly we will do a follow up study.
Other comments are that the technology used (Affymetrix microarrays) is vastly outdated.
Unfortunately, Affymetrix microarrays was the only available tool to assess genes expression at our university.
the healthy control population is not described in any detail (number of participants, male/female, standard blood parameters);
The number of healthy participants was mentioned on methodology line 227-228. The details were not mentioned as all their parameters were withing the normal ranges and the focus of the steady was on SCD patients during both statuses.
the analysis of RUNX3 analysis by ELISA is not convincing and has not been described in sufficient detail. Which antibody was used for detection of RUNX3? Why were the results not validated by western blot analysis which would provide important additional information such as the size of the protein and potentially the presence of isoforms?
"We appreciate the reviewers' comments regarding the ELISA analysis. In our study, we used the Human RUNX3 ELISA Kit from MyBioSource (Catalog Number: Cat. No: MBS9332568 which employs a HRP-Conjugated Human RUNX3 detection antibody for RUNX3 detection. We acknowledge the limitation of not validating our findings through Western blot analysis, which would indeed provide additional valuable information about protein size and potential isoforms. This represents an important future direction for our research to further validate and expand upon our current findings."

Reviewer 2 Report
Comments and Suggestions for Authors
This manuscript explores the potential of transcription factor RUNX3 to serve as a marker of vaso-occlusive crises in sickle cell disease. The manuscript has novel information with clinical relevance. The authors should be praised for clearly writing the methodology used (detailed descriptions of sample collection, RNA extraction, and statistical analysis), as well as for the integration of microarray, qRT-PCR, and ELISA assays to strengthen the findings. Nonetheless I have some comments as follow:
1) The introduction provides a good overview of SCD and VOC but could better highlight the gap in knowledge regarding genetic markers for VOC. Moreover, the authors should remove the last part of the introduction that concerns the MDPI template.
2) The authors should clarify the rationale for focusing on whole-blood transcriptomics over specific cell types.
3) While the downregulation of RUNX3 is well-supported, the authors could include a discussion on why RUNX3 was prioritized over other downregulated genes (e.g., functional relevance, fold change magnitude). Moreover, the discussion of RUNX3 role in immune regulation is thorough, but more detail on how its downregulation specifically contributes to VOC pathophysiology would be valuable. Maybe the authors could consider adding a schematic summarizing the hypothesized role of RUNX3 in VOC.
5) It's important that the authors acknowledge limitations such as sample size, and the need for functional studies to confirm RUNX3's role.
Comments on the Quality of English LanguageThe manuscript could benefit from english proofreading, and correction of grammatical errors.
Author Response
Reviewer 2
This manuscript explores the potential of transcription factor RUNX3 to serve as a marker of vaso-occlusive crises in sickle cell disease. The manuscript has novel information with clinical relevance. The authors should be praised for clearly writing the methodology used (detailed descriptions of sample collection, RNA extraction, and statistical analysis), as well as for the integration of microarray, qRT-PCR, and ELISA assays to strengthen the findings. Nonetheless I have some comments as follow:
Thank you very much for your time in reviewing our work. Your comments and advice are valuable and helpful in improving our paper.
1) The introduction provides a good overview of SCD and VOC but could better highlight the gap in knowledge regarding genetic markers for VOC. Moreover, the authors should remove the last part of the introduction that concerns the MDPI template.
Added line 53-57
2) The authors should clarify the rationale for focusing on whole-blood transcriptomics over specific cell types.
Although many differential gene expression profiling studies have used red blood cells as a source of RNA, recent studies have shown that whole blood RNA is excellent for producing gene expression data with minimal variability and good sensitivity. https://aacijournal.biomedcentral.com/articles/10.1186/s13223-019-0382-x
In our study, we investigated whole blood to evaluate the complete picture of gene expression level in SCD and to build a database regardless of cell origin. The whole blood makes a rich source of easily accessible gene expression data for analysis of physiological or pathological conditions, even in inaccessible tissues. https://www.science.org/doi/10.1126/sciadv.abd6991
Our second objective was to correlate our findings with hemolysis in SCD and we searched for a gene related only to the VOC.
3) While the downregulation of RUNX3 is well-supported, the authors could include a discussion on why RUNX3 was prioritized over other downregulated genes (e.g., functional relevance, fold change magnitude). Moreover, the discussion of RUNX3 role in immune regulation is thorough, but more detail on how its downregulation specifically contributes to VOC pathophysiology would be valuable. Maybe the authors could consider adding a schematic summarizing the hypothesized role of RUNX3 in VOC.
"We thank the reviewers for their insightful comments regarding RUNX3 prioritization and its role in VOC pathophysiology. RUNX3 was prioritized among the downregulated genes based on several key factors:
Molecular Mechanisms of RUNX3 in VOC Pathophysiology:
Immune System Regulation: Based on our GO term enrichment analysis (Figure 4), RUNX3 downregulation significantly impacts immune system processes, particularly through: a) T-Cell Development and Function: RUNX3's role in CD8+ T cell development aligns with our finding of downregulated T cell receptor binding pathways
This is supported by previous studies showing RUNX3's critical function in T-cell lineage specification [32]. The significant downregulation of IL7R (12.76-fold) and CD3E (11.28-fold) in our study further supports this immune dysregulation pathway
Inflammatory Response: Our findings showed significant alterations in inflammatory pathways, specifically: a) Cytokine Regulation: RUNX3 downregulation correlates with altered inflammatory mediator expression. This is evidenced by the substantial downregulation of CCL5 (9.24-fold) in our VOC samples. Previous studies have shown RUNX3's protective role against inflammatory response through regulation of JAK2/STAT3 pathway [35].
Cellular Response Mechanisms: The GO term enrichment analysis revealed: a) Signal Transduction: Upregulation of cellular localization and signal transduction regulation pathways. These changes suggest adaptive responses to the inflammatory and hypoxic environment during VOC.
A detailed figure showing these molecular pathways based on our g:Profiler results:
Integration with VOC Pathophysiology:
The molecular changes we observed suggest that RUNX3 downregulation contributes to VOC through:
- a) Immune System Disruption: Impaired T-cell development and function, evidenced by downregulation of key T-cell markers (IL7R, CD3E). Altered MHC protein complex binding, affecting immune response regulation
- b) Inflammatory Cascade: Disrupted cytokine production and signaling. Altered inflammatory mediator expression (CCL5 downregulation).
- c) Cellular Stress Response: Enhanced cellular localization and signal transduction
Increased ubiquitin ligase complex activity, suggesting altered protein homeostasis
These molecular mechanisms provide a comprehensive framework for understanding how RUNX3 downregulation contributes to VOC pathophysiology and supports its potential as a biomarker for VOC in SCD patients."
The GO terms enrichment analysis shown in Figure 4 from the manuscript to create a more precise molecular mechanism discussion:
Molecular Mechanisms Based on GO Term Enrichment Analysis:
Upregulated Pathways During VOC: From our GO term analysis (Figure 4), three major upregulated processes were identified: a) Cellular Localization: Enhanced protein localization to cell periphery and increased membrane protein localization. These changes suggest active cellular reorganization during VOC. b) Signal Transduction Regulation: Increased regulation of signal transduction and enhanced cellular response to stimuli. These pathways indicate adaptive responses to VOC conditions
- c) Ubiquitin Ligase Complex: Upregulation of ubiquitin protein ligase binding and enhanced protein modification processes. This Suggested increased protein turnover and cellular stress response
A detailed visualization of these GO terms
5) It's important that the authors acknowledge limitations such as sample size, and the need for functional studies to confirm RUNX3's role.
True, as mentioned on line 220 and 221 that the result needs to be validated on large sample size and to study the RUNX3 pathway to assess its functional role on disease pathophysiology. But according to your advice a clear statement was added line 223-224.

Round 2
Reviewer 2 Report
Comments and Suggestions for Authors
The authors addressed my comments and I believe their manuscript can be accepted in its current form.
Author Response
Comments and Suggestions for Authors
The authors addressed my comments and I believe their manuscript can be accepted in its current form.